# Geomorphological Signature of Late Pleistocene Sea Level Oscillations in Torre Guaceto Marine Protected Area (Adriatic Sea, SE Italy)

**Francesco De Giosa [1], Giovanni Scardino [2] , Matteo Vacchi [3,4] , Arcangelo Piscitelli [1], Maurilio Milella [1], Alessandro Ciccolella [5] and Giuseppe Mastronuzzi [2,\*]**

[1]  Environmental Surveys Srl, Via Dario Lupo 65, 74121 Taranto, Italy; francescodegiosa@ensu.it (F.D.G.); arcangelo.piscitelli@libero.it (A.P.); maurisismo@libero.it (M.M.)
[2]  Dipartimento di Scienze della Terra e Geoambientali, Università degli Studi di Bari "Aldo Moro", Via Edoardo Orabona 4, 70125 Bari, Italy; giovanni.scardino@uniba.it
[3]  Dipartimento di Scienze della Terra, Università di Pisa, Via Santa Maria 53, 56126 Pisa, Italy; matteo.vacchi@unipi.it
[4]  CIRSEC, Center for Climate Change Impact, University of Pisa, Via del Borghetto 80, 56124 Pisa, Italy
[5]  Consorzio di Gestione di Torre Guaceto, Via Sant'Anna 6, 72012 Carovigno (Brindisi), Italy; segreteria@riservaditorreguaceto.it
\*  Correspondence: giuseppe.mastronuzzi@uniba.it

**Abstract:** Morphostratigraphy is a useful tool to reconstruct the sequence of processes responsible for shaping the landscape. In marine and coastal areas, where landforms are only seldom directly recognizable given the difficulty to have eyewitness of sea-floor features, it is possible to correlate geomorphological data derived from indirect surveys (marine geophysics and remote sensing) with data obtained from direct ones performed on-land or by scuba divers. In this paper, remote sensing techniques and spectral images allowed high-resolution reconstruction of both morpho-topography and morpho-bathymetry of the Torre Guaceto Marine Protected Area (Italy). These data were used to infer the sequence of climatic phases and processes responsible for coastal and marine landscape shaping. Our data show a number of relict submerged surfaces corresponding to distinct phases of erosional/depositional processes triggered by the late-Quaternary interglacial–glacial cycles. In particular, we observed the presence of submerged marine terraces, likely formed during MIS 5–MIS 3 relative highstand phases. These geomorphic features, found at depths of ~26–30, ~34–38, and ~45–56 m, represent important evidence of past sea-level variations.

**Keywords:** morphostratigraphy; sea-level changes; marine terraces; river incisions; Adriatic Sea

## 1. Introduction

Ice-cores and marine sediment records indicate that the climate of the last 500 ky was characterized by ~100 ky warm–cold cyclicity [1,2] which led to repeated transitions between glacial and interglacial periods. These transitions triggered cycles of accretion and melting of the major ice-sheets with consequent major oscillations of sea-level position [3–5] and significant modifications of the on-land and sea-floor landscapes.

Morphostratigraphy applied to landscape evolution has allowed recognition of relict sequences of past morphogenetic processes associated with these glacial and interglacial climatic phases [6,7]. In particular, past landscapes have been often reconstructed through a combination of geomorphological and sedimentological analyses [8–10]. In the Mediterranean Sea, this multidisciplinary approach has proved to be very useful to understand genesis and evolution of particular landforms and sea-floor

features as well as to infer relative sea-level (RSL) changes and their influence on coastal landscape evolution (e.g., [11–14]).

However, many relict landforms are often difficult to observe and to analyze because they are located in submarine areas or covered by thick layers of more recent sediments. This is, for instance, the case of colder climatic phases when sea level was many m below the modern position, considering the tectonic contribution as well (e.g., [15,16]). This issue was progressively resolved thanks to recent technological advances that allow collecting high-resolution data to characterize the morpho-bathymetry and morpho-topography of a specific zone, even in underwater environments [17–19], using remote sensing techniques and spectral images.

In this paper, we investigated the coastal and off-shore zones of the Marine Protected Area (MPA) of Torre Guaceto (Adriatic Sea, SE Italy, Figure 1) through remote sensing techniques, morphological, and stratigraphic surveys in order to recognize and correlate chronologically subaerial and submerged landforms.

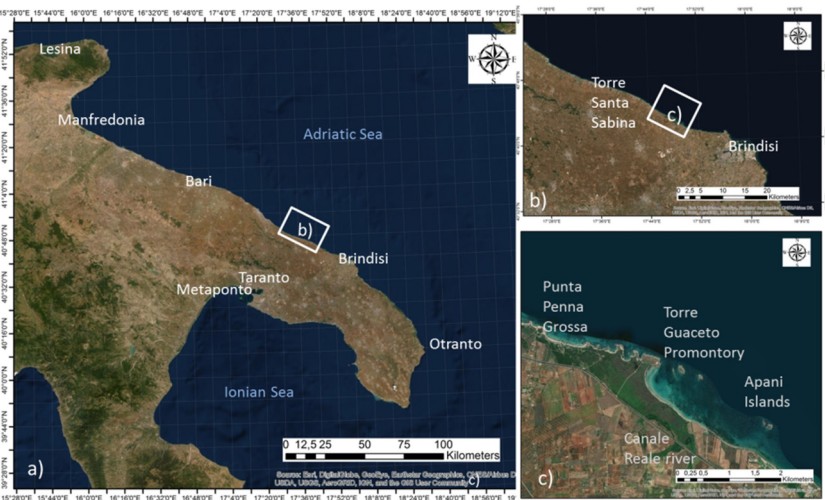

**Figure 1.** (**a**) Study area located in Apulia region; (**b**) location of Torre Guaceto surveyed area (Adriatic Sea, SE Italy); (**c**) surveys were performed from Punta Penna Grossa to Apani Islands.

Pleistocene marine terraces and Holocene deposits can provide an important insight into late-Quaternary rates of vertical displacements [20,21]. In south-eastern Apulia, evidence of Marine Isotope Stage (MIS) 5 are rare and poorly constrained in terms of chronology and elevation. The sole available indirect age comes from the south-eastern Murge (Torre Santa Sabina locality near Brindisi, Figure 1b). Here, a coastal deposit situated at ~3 m a.s.l. overlies a colluvial deposit bearing Late Paleolithic–Mousterian flints and, thus, it can be correlated to a generic MIS 5 [22,23]. In the northern part of Apulia, near Manfredonia (Figure 1a,b), MIS 5 deposits were found at depth of −22 m [24]; it implies significant subsidence rate (−0.17 mm/y) that are comparable to the areas placed in a very different geodynamic context (e.g., Trieste, Versilia, and Sarno plains [20]).

In this study, we identified the presence of relict marine terraces and other subaerial landforms presently lying along the Apulian continental shelf. We correlated them to the major late-Quaternary climatic phases coupling models with data available on land [20,21]. This analysis allowed reconstructing the coastal evolution of the Torre Guaceto Marine Protected Area (MPA) during the last 150 ky.

## 2. Geological Settings

Torre Guaceto is a tower of the XV century, placed on a promontory located north of the town of Brindisi, inside of the MPA of Torre Guaceto (Figure 1). The study area overlaps the Canale Reale river, which divides the Murge plateau from the Taranto-Brindisi plain [25,26]. Different lithological units crop out in this area, whose deposition is connected to past sea level stands and tectonic factors

during Middle-Late Pleistocene [25,27]. The bedrock is represented by Mesozoic limestone, which is overlain by discontinuous marine deposits of Plio-Pleistocene age (up to 70 m), belonging to the Calcarenite di Gravina Fm, and the etheropic argille subappennine informal unit [28,29]. These units are covered by middle-upper Pleistocene biocalcarenitic beach and dune deposits [11,25,30]. Along the Murgia scarp they crop out as stepped terraces stretching from −400 m to a few m above the present mean sea-level. Although along the coastal area of Ionian Apulia the younger marine terraces are generally characterized by the presence of a senegalensis faunal assemblage that, in combination with U/Th ages, indicates its deposition during the MIS 5.5 (e.g., [23,31–33]), the Adriatic coastal area of Apulia does not permit any chronological attribution due to the lack of geochronological or paleontological markers. Every chronological attribution related to the upper/late Pleistocene derives from the use and applications of the morphostratigraphy principles [6]. All along the Adriatic coast stretching north to Brindisi and in particular between Torre Guaceto and Punta Penne, sea-level drop associated with the last glacial maximum (LGM, ~26 to ~21 ky BP, [34]), caused the incision of the basement, in correspondence on the current hydrographic network, showing features of sapping processes [11,23]. Seaward, sapping valleys cut the upper Pleistocene biocalcarenitic sedimentary cover. Surveys performed on the land allowed forming a hypothesis that the shaping of the paleo-beach-dune system occurred during MIS 5; the sapping valleys were shaped due to the increase of the relief energy caused by the lowering of sea level. Their maximum shaping would correspond to the lowermost sea-level stand (−120 m) at the LGM [11,25,27,30].

The following Holocene marine transgression (last 12 ky BP) and inter-strata dissolution produced a series of sub-circular inlets that host pocket beaches (Figure 2). On their border, a polyphasic dune ridge was recognized, formed by two aeolian sediment generations, the first at ~6.0 ky BP and the second at ~2.5 ky BP [11,25,30]. The south-eastern part of the foredune extends for ~500 m reaching a maximum elevation of −12 m msl. This is composed of brownish sand layers intercalated with brown soil rich in *Helix* spp. [30,35]. Geo-archaeological and bio-stratigraphic analysis indicate that RSL was 2.25 ± 0.2 m below the present one at about 3.5 ky BP and that the total RSL variation in the last ~2.0 ky BP was about of 0.9 m below the present mean sea-level (msl) [36,37].

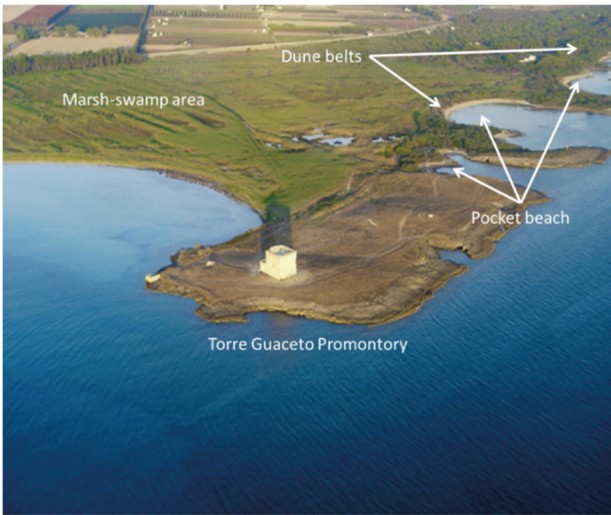

**Figure 2.** The promontory of Torre Guaceto is shaped on a sequence of Calcarenite di Gravina Fm. (late Pliocene–Early Pleistocene) and of late Pleistocene–Tyrrhenian biocalcarenite [26].

The comparison of these data with the available GIA (glacial isostatic adjustment) models [38,39] indicates a low rate of tectonic subsidence of this coastal area at least during the last 125 ky. This is further corroborated by the absence of significant historical seismicity and by the GPS data that indicate zero to weakly negative on-going vertical movements [40]. Subsidence rate increases on the northern sector of Apulia, reaching values of −0.3 mm/y as indicated by tectonic structures observed in

seismic profiles and boreholes off-shore the Gargano Promontory and Tremiti Islands [41,42]. In this tectonic framework the significant sea-level changes are added, in particular during the last 100 ky, when sea-level falls on MIS 5.4 and MIS 5.2 determined downward and seaward shifts of the shoreline and the decrease of sediment supply, possibly in response to the reduction in basin width that hampered lateral advection [43].

## 3. Materials and Methods

In this work, we merged high-resolution bathymetry derived by LIDAR remote sensing (with a penetration into water-column of ~50 m) techniques with detailed MIVIS spectral images (with a penetration into water-column of −7 m). We further corroborate these data by scuba diving surveys (e.g., [44,45]).

Remote sensing data consisted LIDAR data (Airborne Laser Terrain Mapper—ALTM Optech's Gemini 167 kHz, near infrared, Teledyne Optech, Toronto, ON, Canada) and Daedalus AA5000 MIVIS (Multispectral Infrared and Visible Imaging Spectrometer, Italian National Council Research CNR, Italy) hyperspectral images both for the on-shore and nearshore coastal zone (e.g., [17]). LIDAR Optech's Gemini 167 data are part of a wider acquisition of remote sensing data in the areas ascribed to the protected marine areas of Calabria, Campania, Apulia, and Sicily. This LIDAR system allows obtaining elevation data to an accuracy of 5 to 10 cm. This LIDAR system flies up to 4000 m to cover large coastal area where a high degree of accuracy and speed is necessary and where accessibility is difficult, as in back-dune zones or in steep coastal slope.

The surveys collected a point distribution, with each point consisting of x-y-z coordinates and associated reflectance value. Acquired points were processed in GIS environment to build digital surface models (DSMs) and digital terrain models (DTMs). In order to characterize the submerged coastal zone particularly for the near shore bathymetry up to −7 m, hyperspectral images have been used and acquired with MIVIS scanner. This instrument, property of the National Research Council of Italy, is a 102 channel scanner covering visible and near infrared (0.43–0.83 μm), middle infrared (1.15–1.55 and 1.98–2.50 μm) and thermal infrared (8.21–12.70 μm) regions of the electromagnetic spectrum, providing a wealth qualitative information of the surveyed area. MIVIS scanner has a geometrically correct scan line that, due to the movement of the aircraft, is displaced with the roll, pitch, yaw, and with changes in velocity and direction. The operational flight heights of the scanner can range from 1500 up to 5000 m above ground; at this height, the nadiral pixel dimension ranges from 3 up to 10 m, integrated with a GPS system and a gyro.

In GIS environment, LIDAR data and hyperspectral images have been combined to build digital terrain model (DTM) and digital surface model (DSM) with a grid cell width of 4 × 4 m, in order to define the major landforms occurring both along the coast and the shallow continental shelf up to a depth of −56 m (Figure 3).

Two scuba diving surveys across the incision immediately to the ESE of the promontory of Torre Guaceto (Figure 4) have been performed. The geomorphological surveys were traced up to a depth of about 18 m along both sides of the incision (Figure 5).

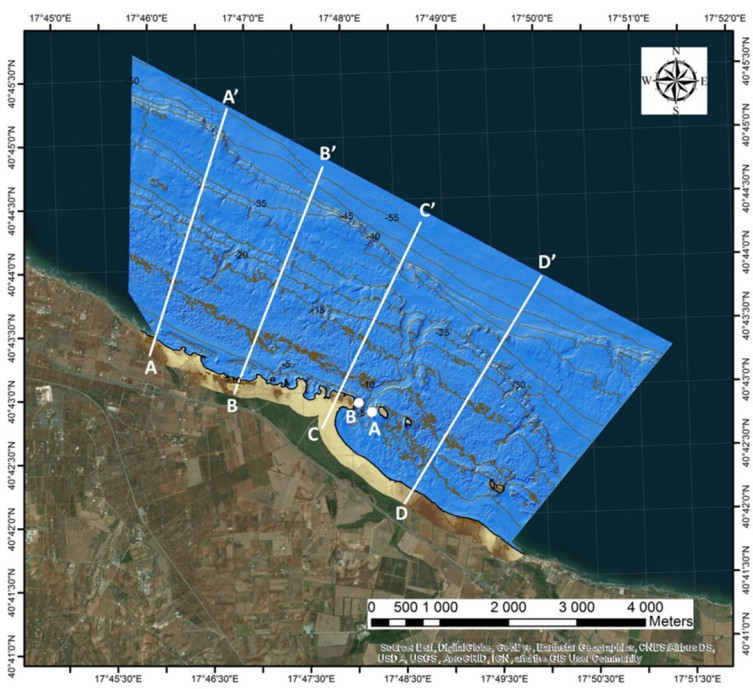

**Figure 3.** Morpho-topographic and morpho-bathymetric DTM (in blue) and DSM (in brown) of Torre Guaceto area with bathymetric profiles traces (in white) and direct scuba surveyed areas (in white dots).

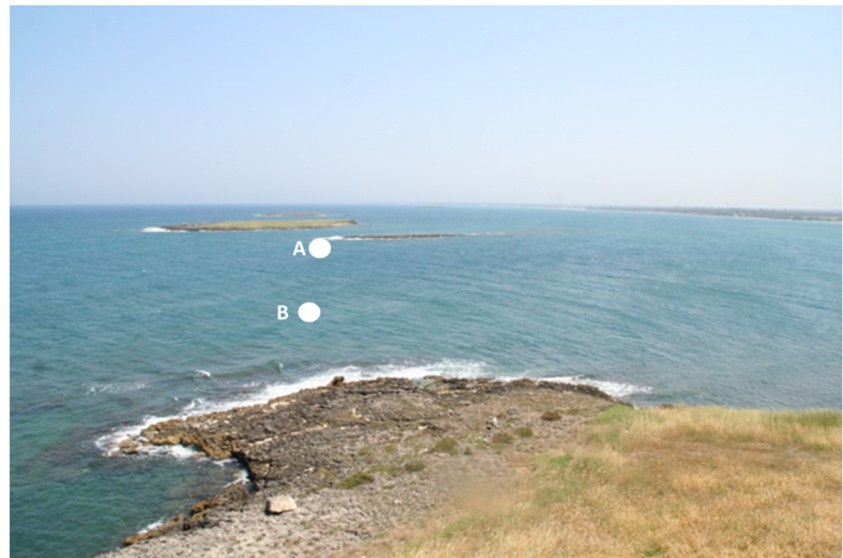

**Figure 4.** The channel between Torre Guaceto promontory (reported in aerial view of Figure 3) and homonymous islands (in background); the sea-floor is cut by a sapping valley characterized by classic box profile, shaped in the Calcarenite di Gravina Fm. up to a depth of about 18 m.

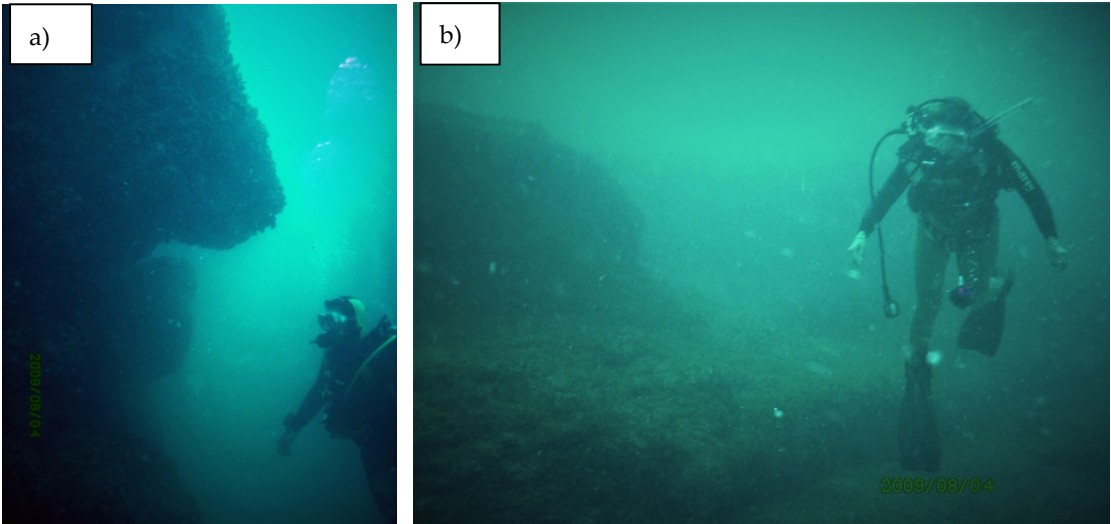

**Figure 5.** Sapping notch and unstable block (**a**) and two large sub-horizontal steps (**b**) shaped on the southern slope of the submerged sapping valley between the islands and Torre Guaceto Promontory.

## 4. Results

The coastal tract comprised between Torre Guaceto and Punta Penne (Figure 1b), is characterized by a sequence of small calcarenitic islets (Apani Islands). They are made of cemented beach and dune sediments which likely represent the MIS 5.5 deposits according to the correlation with similar deposits outcropping all along the coast of Apulia. In addition, a further continuous dune belt can be found inland [23,25,31,32].

The composite survey campaign, performed in this study, allowed recognizing a number of significant submarine landforms; in particular, we observed the presence of near-flat surfaces and other morphological features likely related to submerged incisions. Bathymetric reconstruction revealed a staircase of near-flat surfaces which can be observed all over the different surveyed sectors (Figure 6).

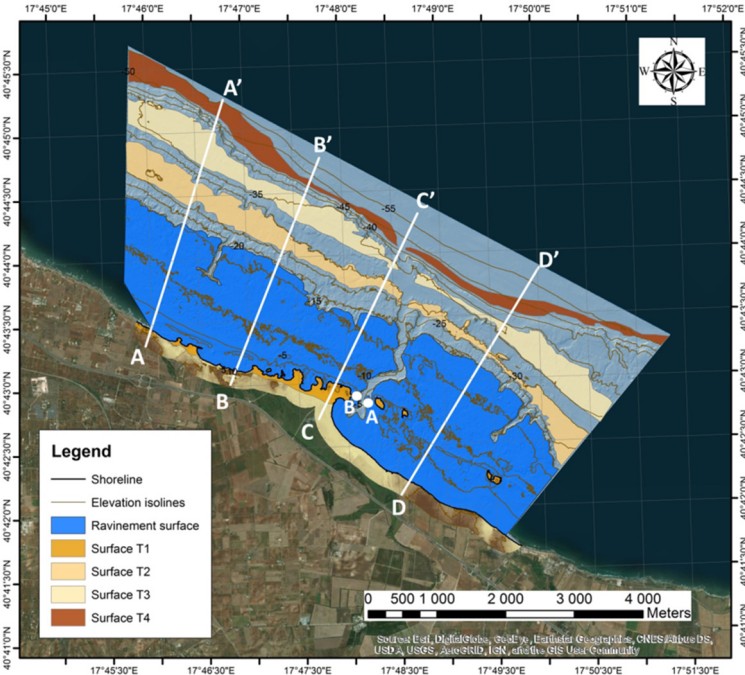

**Figure 6.** Staircase geometry of near-flat surfaces and ravinement surface observed along the Torre Guaceto shelf.

These surfaces are characterized by gentle slopes which do not exceed 4–5 degrees (about 6%) occurring at four different depth ranges (represented in T1-T2-T3-T4 levels in Figure 7) in all the analyzed transects (A, B, C, and D).

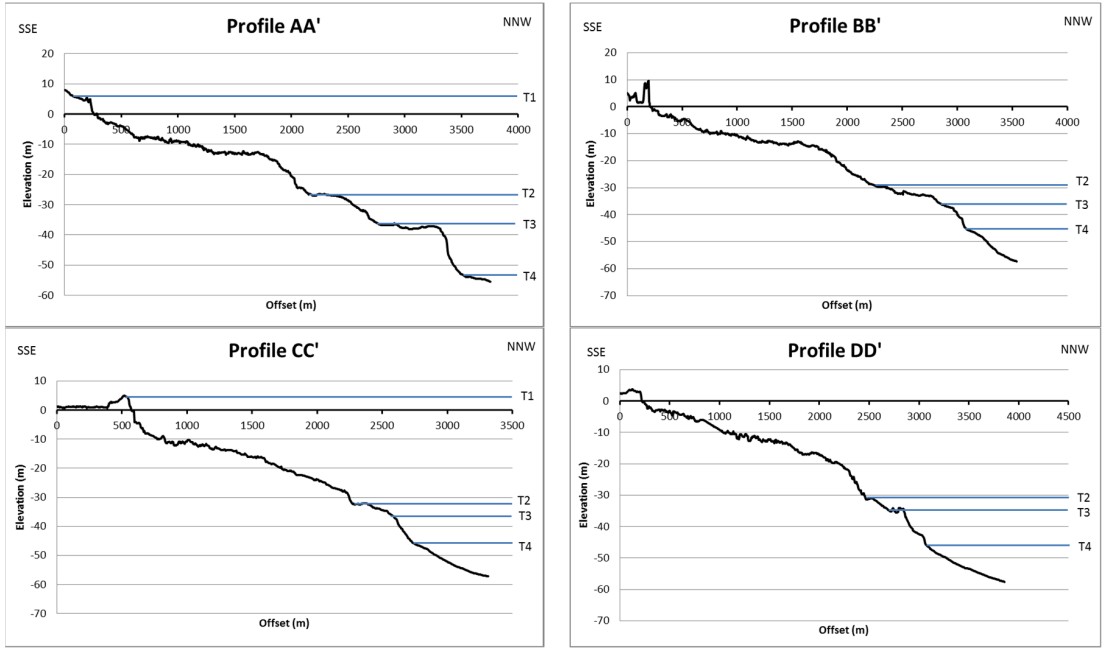

**Figure 7.** Elevation profiles of Torre Guaceto MPA continental shelf. In blue, the upper limit of the surfaces (T1-T2-T3-T4) connected to the past sea-level stands are represented.

The T1 level occurs on shore; its inner (landward) and outer edge (seaward) are placed at 10 and 5 m, respectively.

The second surface (T2 level) occurs underwater. It develops between −26 (inner edge) and −32 m (outer edge). A third erosive surface (T3 level) occurs between −34 (inner edge) and −38 m (outer edge) while a fourth erosional surface (T4 level) ranges between −45 (inner edge) and −56 m (outer edge), with a great extent in correspondence of the profile AA' (Figure 7).

Our scuba surveys further documented a ravinement surface. This surface is gently sloping (2–3°) seaward, developing between the present shoreline and −18 m (Figure 8). It is discontinuously covered by coarse and medium sands which, in some sheltered areas, define the present beaches [46].

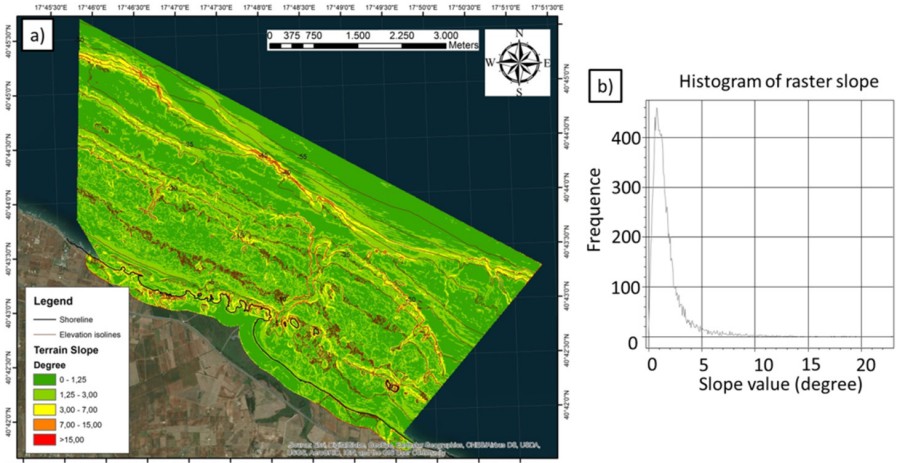

**Figure 8.** Slope analysis of Torre Guaceto continental shelf; (**a**) slope changes in correspondence of the near-flat surfaces limits; (**b**) the mean morpho-bathymetry slope.

Both bathymetric data and scuba diving surveys showed the presence of submerged incisions orthogonal to surfaces boundaries and connected to the current land hydrographic network. These sea-floor features, recognizable along the whole investigated portion of the sea-bottom (e.g., from 0 to −56 m), cut all the submerged near-flat surfaces (T2 to T4, Figure 7).

## 5. Discussion

The new sets of topographic and bathymetric data reveal well preserved evidence of past sea-level stands that shaped the coastal and marine landscape near Torre Guaceto. Unfortunately, the geomorphological markers described in this study lack dateable material; therefore, the chronological constraint of their origin can be only speculated on the basis of bathymetric cross-correlations. In the first approximation, our geomorphological markers can be compared with modelled eustatic values [45,47]. The remains of the highest terrace (T1), most likely formed during the MIS 5.5 (~125 ky) when the RSL was −7 m above the present msl [20,23,37,48].

According to Rovere et al. [49] marine terraces can be shaped by marine erosion or can consist of shallow water to slightly emerged accumulations of materials redistributed by shore erosional and depositional processes (e.g., marine-built terraces) [50]. The width of marine terraces ranges from few hundreds of meters to up to 1–2 km and can stretch along many kilometers of coastline. The mapped submerged near-flat surfaces fit well with this morphological description. For this reason, we interpreted the surfaces as relicts of marine terraces. In the absence of any evidence of discontinuity in the sedimentary bodies, it is impossible to find any evidence of their depositional or erosive genesis. However, the hypothesis that these submerged surfaces could represent erosional marine terraces seems to be supported by seismic profiles performed in central Adriatic Sea, where a significant erosion of MIS 5.5–5.1 shelf progradational units during last 100 ky was observed [26,27].

The chronological frame of the submerged surfaces is complex. This is mainly because the presence of submerged features is very seldom reported in the Mediterranean [44]. The sea-level stands that shaped the reconstructed terraces must be located below both the present and the last interglacial ones. According to the available eustatic curves, different Marine Isotope Stages (3, 5.1, 5.3, 6.5, 7.1, 7.3, and 7.5) peaked below both MIS 5.5 and MIS 1 sea-levels. However, the evidence of river incisions shaped through sapping processes [11,23] is observed in all the submerged marine terraces. This incision reached maximum rate during LGM [6], when sea level was −120 m lower. For this reason, all surfaces must necessarily be older than MIS 1 and younger than MIS 5.5.

The submerged terrace found at depths −26 and −32 m (T2 level) can be tentatively attributed to MIS 5.3 (~101 ky BP) that peaked at −30 m on the sea-level curve by Grant et al., 2014 [5]. At MIS 5.1 (~81.5 ky), sea-level stand allowed the genesis of a new erosional marine terrace encountered at a depth variable between −34 and −38 m, corresponding to T3 level at −38 m. In the following phase, a sea-level drop was observed up to a new sea-level stand on MIS 3 at ~54.5 ky, with formation of marine terrace in correspondence of T4 level, at a depth variable between −45 and −56 m, peaked at −52 m on the sea level curve by Grant et al., 2014 [5].

The general tectonic framework of Torre Guaceto area reveals a low subsidence rate of 0.02 mm/y [15,22,23,51]. We corrected the current depth of terraces according to the subsidence rate in order to attribute the actual elevation for each boundary at the moment of their genesis (Table 1).

In the case of the MIS 5.5 surface, a displacement of 2.5 m in 122 ky has been calculated considering a tectonic rate of 0.02 mm/y [22,37,48]. This implies a corrected surface altitude ranging between 12 (inner edge) and 3 m (outer edge) at the moment of surface genesis.

For the MIS 5.3 surface, a displacement of 2.02 m has been calculated in 101 ky. It implies a depth range between −23.98 (inner edge) and −28.98 m (outer edge), while for the MIS 5.1 surface a displacement of 1.63 m has been calculated in 81.5 ky attributing a corrected depth range between −32.37 (inner edge) and −36.37 m (outer edge). Finally, for the MIS 3 surface, a displacement of 1.09 m has been calculated in 54.5 ky which implies a depth range between −43.91 (inner edge) and −54.91 m (outer edge) at the moment of surface genesis.

**Table 1.** Depth of surfaces detected in each profile corrected for the tectonic displacements.

| Surface | Depth Range Profile AA' (m) | Depth Range Profile BB' (m) | Depth Range Profile CC' (m) | Depth Range Profile DD' (m) | Sea Level Highstand SPECMAP Imbrie & McIntyre 2006 | Sea Level Highstand Walbroeck et al., 2002 | Sea Level Highstand Grant et al., 2014 |
|---|---|---|---|---|---|---|---|
| MIS 5.5_T1 | 6.26 and 4.17 | - | 5.76 and 3.73 | - | 0.34 m at 122 ky | 6.32 m at 123.87 ky | 10.62 m at 122 ky |
| MIS 5.3_T2 | −24.57 and −25.11 | −27.05 and −31.23 | −29.92 and −31.41 | −29.02 and −29.6 | −35.58 m at 99 ky | −20.86 m at 101 ky | −30 m at 101 ky |
| MIS 5.1_T3 | −34.42 and −36.27 | −34.32 and −36.79 | −34.74 and −36.06 | −33.38 and −33.75 | −35.21 m at 80 ky | −18.67 m at 81.5 ky | −38 m at 81.5 ky |
| MIS 3_T4 | −52.43 and −54.11 | −44.6 and −47.11 | −44.71 and −46.53 | −44.28 and −46.82 | −76.79 m at 54 ky | −52.08 m at 54.5 ky | −52 m at 54.5 ky |
| Ravinement surface | −2.8 and −12.92 | −3 and −13 | −7.7 and -16 | −3.3 and −13.7 | −26.27 at 7 ky BP | −9.74 m at 8 ky BP | −2.36 m at 7 ky BP |

These terraces can be referred to different relative past sea-level stands occurring during MIS 5-3, corresponding to peaks which can be observed in different model curves (e.g., [5,52–55] Figure 9).

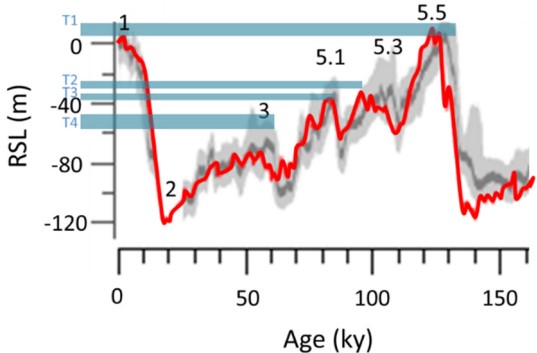

**Figure 9.** Sea-level changes from 150 ky to the present derived from Red Sea records (modified after Grant et al., 2014). Light blue bands show erosional marine terraces depth ranges surveyed in the Torre Guaceto area.

The position of each marine terrace fits well with evidence of MIS 5 deposits already described by Mastronuzzi et al., 2011; 2018 [25,26]. This correlation was made under the assumption of minimal tectonic movements of this area [52] that show significantly different neotectonics pattern with respect to the northern part of the Apulia [21,24,34–36].

Morphobathymetry suggests that a valley network developed coeval with the sea-level stand through sapping processes [11,23]. In fact, since the evolution of the aquifer and the seawater/fresh water interface is strictly linked to the sea-level, the development of each sapping valley was largely influenced by the Late-Quaternary sea-level changes and the consequent shifts of the coastline [11,30]. Each sea-level (high) stand induced the development of a marine terrace, a shoreline and a number of short valleys. These valleys developed orthogonally to the coastline since the sapping processes was conditioned by structural alignment and/or by general geometry of the local basement [11]. Maximum incision occurred during MIS 2, when sea level was 120 m lower than present.

During the Holocene sea-level rise, incisions were flooded and filled by sediments, until the slowing down of sea-level rising rates (7 ky BP, [38,56], Figure 10), causing the widening of the ravinement surface between −18 m and the present sea level.

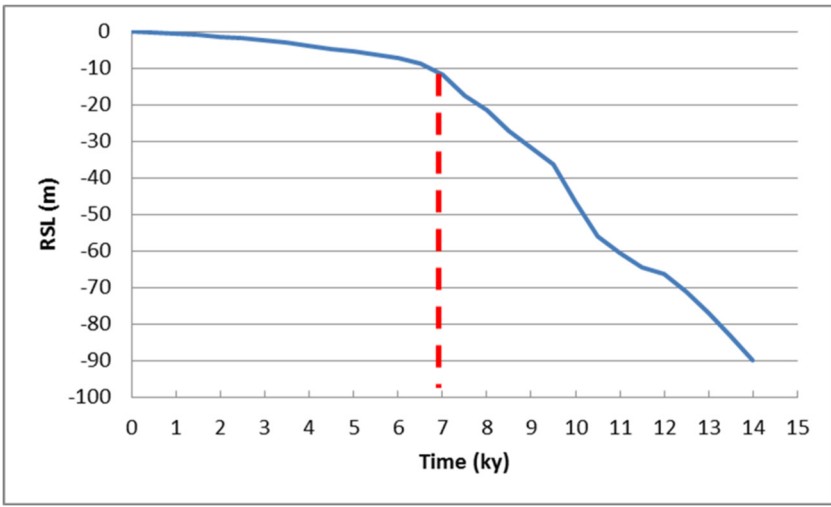

**Figure 10.** Relative sea-level prediction for Egnatia site [41]; the red dashed line indicates the slowing down of the sea-level rising rate at 7 ky.

Applying the morphostratigraphy principles on the near-flat surfaces depth—both subaerial and submerged—and to the sea-level trend recognizable during the last warming time, it is possible to reconstruct the following morpho-evolutionary steps:

- MIS 5.5—the sea-level highstand at 7 m msl allowed the deposition of beach and dune deposits shaping the marine terrace corresponding to the T1 level;
- MIS 5.3—during the general regression, the relative sea-level stand shaped the marine terrace currently located at a depth range between −26 and −32 m msl (T2 level);
- MIS 5.1—the relative sea-level stand allowed the shaping of the marine terrace currently located between −34 and −38 m msl and corresponding to the T3 level;
- MIS 3—the relative sea-level stand induced the marine terrace shaping currently located between −45 and −56 m msl (T4 level);
- MIS 2—at the LGM, the sea-level placed at −120 m msl facilitate the full incision on valley network;
- MIS 2—the post-LGM sea-level rose at fast rates until 7500 years BP;
- 7000 years BP—the sudden slow in sea-level rising rates produced the widening of the ravinement surface surveyed between −18 m msl and the present msl;
- 3500 years BP—sea-level stands during the Bronze Age (3.5 ky BP) at about −2.25 ± 0.2 m msl;
- 2200 years BP—sea-level stands at about −1.1 ± 0.1 m msl below the present mean sea level; inlets were marked by the presence of beaches with low embryonic dunes and typical back-beach environments;
- 1900 years BP—sea-level stands at about −0.65 ± 0.1 m msl;
- 1700 years BP—sea-level probably stands around −0.3 ± 0.1 m msl

## 6. Conclusions

Morphostratigraphic approach carried out in Torre Guaceto area allowed recognizing and chronologically correlating a variety of sea-floor features shaped by different sea-level stands in the Late-Quaternary. Last interglacial phases were already described in the literature [25,26,48] while, for the submerged features, new technologies have been useful to individuate the morphodynamic phases following the MIS 5.5 sea-level highstand.

Remote sensing and spectral images allowed detecting the underwater landforms which were used to perform a high-resolution mapping of submarine environments at large spatial scale. In particular, the use of LIDAR and MIVIS instruments allows surveying up to a depth of 55 m, highlighting the main underwater landforms as near-flat surfaces and submerged fluvial incisions.

Morphostratigraphic analysis of the sea-floor features allowed reconstructing the different shaping phases that were correlated to the chronological constrains deriving from the study of the shallow water and subaerial landforms. This analysis clearly indicated that the underwater marine terraces of Torre Guaceto were older than MIS 2; we tentatively attributed their formation in correspondence of the highstand peaks of MIS 5.5–MIS 5.3–MIS 5.1–MIS 3 (Figure 11).

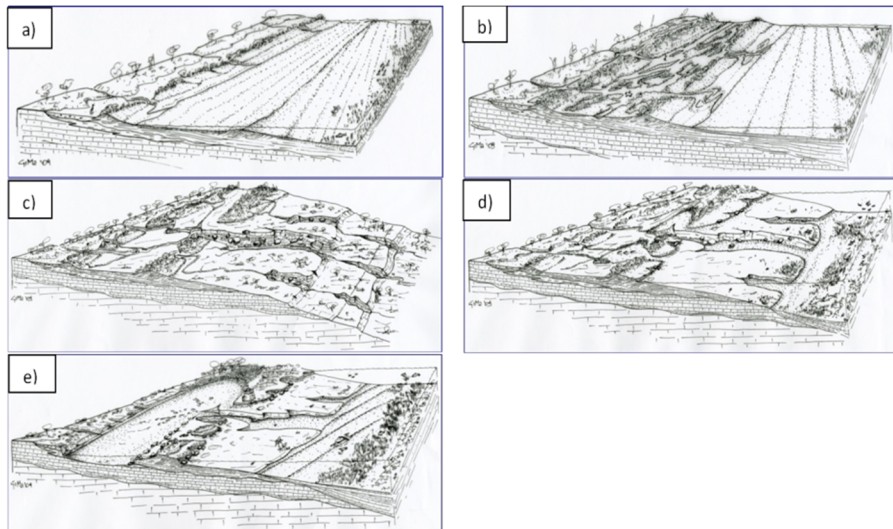

**Figure 11.** Phases on Torre Guaceto area: (**a**) First Marine Isotope Stage 5.5 phase–(**b**) second Marine Isotope Stage 5.5 phase–(**c**) last glacial maximum (LGM)–(**d**) bronze Age–(**e**) present.

The morphostratigraphic approach presented in this paper represents a fundamental first step to investigate the submerged landforms; further investigation, possibly corroborated by additional coring campaign may provide more precise insights into the genesis of the submerged landforms and into the climatic phases that shaped them.

**Author Contributions:** Conceptualization: F.D.G., G.S. and M.V.; Data curation: F.D.G. and M.V.; Formal analysis: M.V.; Funding acquisition: G.M.; Investigation: A.P.; Methodology: M.M.; Project administration: G.M.; Resources: A.C.; Visualization: A.C.; Writing—original draft: F.D.G. and G.S.; Writing—review & editing: A.P., M.M. and G.M.

**Funding:** This research has been realized by a research agreement between the Consortium of Torre Guaceto and the Department of Earth and Geo-environmental Sciences of the University of Bari, approved in the Department Council on 1 March 2019.

**Acknowledgments:** This paper is the result of studies performed in the framework of the agreement between the Torre Guaceto Marine Protected Area and the Department of Earth and Geo-environmental Sciences of the University of Bari. We thank all collaborators for their logistic and technic support in every phase of this work. We are thankful to the reviewers for their revisions that allow us to improve the quality of the paper. MV is funded by the Rita Levi Montalcini programme of the Italian Ministry of University and Research (MIUR).This work has been carried out under the umbrella of the IGCP Project n. 639 "Sea-level change from minutes to millennia" (Project Leaders: S. Engelhart, G. Hoffmann, F. Yu and A. Rosentau). We extend our gratitude to the MOPP-Medflood (INQUA CMP 1603P) project for fruitful discussions during the workshops.

**Conflicts of Interest:** The authors declare no conflict of interest. The funders had no role in the design of the study; in the collection, analyses, or interpretation of data; in the writing of the manuscript, or in the decision to publish the results.

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
