# Peer review of "Geomorphological Signature of Late Pleistocene Sea Level Oscillations in Torre Guaceto Marine Protected Area (Adriatic Sea, SE Italy)"

_water, doi:10.3390/w11112409_

Round 1

Reviewer 1 Report

The paper by De Giosa et al. provides new information on outcropping and submerged landforms, which are interpreted as marine terraces, located in the quasi-stable Adriatic coast of Apulia, in southeastern Italy. The study is based on high-resolution topographical and bathymetrical surveys carried out in the Torre Guaceto area. The authors suggest a correlation, based on geomorphological criteria, of the identified marine terraces with sea level highstands of the Late Pleistocene, and propose a morpho-evolution scheme for the investigated area.

The work by Giosa et al. may contribute to the reconstruction of recent sea level fluctuations in the Mediterranean region, and may represent a valuable contribution to further studies aimed at constraining relative sea level change both in the region investigated with the study at issue and in the entire Mediterranean region, provided some revision mainly to the data presentation and discussion.

In particular, the work would be improved by adding to the figures a map showing the terraces, and by presenting/discussing evidence suggesting that the smooth submerged surfaces that have been identified are erosional surfaces/ravinement surface. I also suggest moving figures 3 to 5 to the results section. 

Overall, I suggest some reorganization of the discussion section (see annotated manuscript) and a more comprehensive discussion of the interpretation of the results. In fact, although a nice review of recent studies on recent global/local sea level positions during recent MIS is shown in Table 1, a discussion on the hypothesised correlation with MIS 5.3, 5.1 and 3 and, particularly, with the peaks in the curve by Grant et al (2014) would improve the work. I suggest the authors to present in the Introduction section a framework of the vertical distribution of MIS5.5 sea level markers in the Adriatic coast of Apulia and of evidence pointing to (pre-Holocene?) subsidence of the study area, according to the literature. Some consideration on possible future developments of this study can be added in the Conclusion section. 

Detailed comments are reported in the attached annotated manuscript.

I have noted several spelling/syntax mistakes (some are evidenced in the annotated file). 

Author Response

Thank you for your revisions.

Reviewer 2 Report

Comments for: Water 602828

De Giosa et al. 2019

     This is a straightforward geomorphic investigation of an important set of potential Late Pleistocene and Holocene sea-level indicators in Italy. It effectively combines modern Lidar and multispectral scanning with scuba-diving examinations for ground truth. Keys to this research are: 1) identification of clear and correlative terraces and notches at distinct levels on this stretch of the Italian coast, 2) relationship of these level indicators to well-established sea-level fluctuations, 3) eliminating or minimizing the effects of tectonic displacement, and 4) establishing a time as well as level correlation.

     Point (1) is best evaluated by comparing Table 1 and Figure 6. Unfortunately, the authors are a bit inconsistent in the ranges of the depths cited, and the terraces are not clearly consistent on the four profiles. In the profiles of Figure 6, I find it hard to see unique terraces clearly, but Figure 3 seems to be better. Examination of Figure 3 shows several features that bifurcate or merge along the coast. For example, the two deepest scarps between A and B merge to a single scarp between B and C. Also, the shelf between C and D is very disrupted by the river sapping incision. The shelf to the ESE of D shows a distinct landward swing of a scarp between -30 and -20 m. For these reasons, I cannot accept Point (1) as fully valid.

   Point (2) is highly dependent on Point (1), but for individual profiles, the correlation to MIS5 and later sea-level highstands seems reasonable.

   As to Point (3), the authors demonstrate by using references and the elevations of the onshore terraces that this region is relatively stable, and are able to make reasonable corrections for tectonic subsidence (Table 1) (although two decimal place precision is a vast overreach).

   Point (4) is difficult, as admitted by the authors, as the carbonate materials are not well suited to radiometric dating. However, stratigraphic placement of the incised Late Pleistocene materials makes the relationship to MIS5 and later the most reasonable inference.

   Unfortunately, the writing is in somewhat imprecise English, and includes numerous misuses of language and punctuation. I have made many edits and comments on the manuscript. In addition, I summarize here some of the more important detailed comments, indexed to line numbers (L).

17 – “seldom directly recognizable” - Nonsense. Explain more precisely. 21-26 – Extremely run-on sentence. Simplify or divide. 28 – “depths of ~26/30 m, ~34/48 m, ~45/56 m.” These should be formatted as 26-30 m, etc. 29 – “Sea-level variations,” “sea-level rise,” but not “rising sea level” are HYPHENATED compound modifiers. Check throughout for similar constructions. 44 – proper punctuation of “e.g.,” “i.e.,” and similar usages always include the comma (they are abbreviations of parenthetical phrases). 46 – This is a better explanation, see comment 1. 54 - "Bedforms" has a specific geologic meaning - the sedimentary structures produced at the base of flow by hydrodynamic processes. I don't think that's what you mean here. “Terraces,” “shelf morphology,” or “sea-floor features” might be better terms. 60 - Figure should be larger: the full width of the column, and the annotations larger to be readable. 62 - "outcrop" is a noun, the proper verb construction is "crop out." I know this is widely violated in the literature, but that doesn't make it right. I note that you use it correctly in line 67. 65 – “the etheropic argille subappennine.” I am unfamiliar with this unit - should it be a formal name (capitalized)? Explain more fully. 75 – “the engravement” is not an English usage – I suggest “incision.” 78 – “allowed to do hypotheses on these two evidence, which can be attributed the” Extremely unclear. Do you mean "allowed forming a hypothesis that the shaping of the paleo-beach-dune system occurred during MIS5 ..." ? 81 – “which was connected to the cooling” is unnecessary here, and strictly incorrect. Cooling is part of a climate signal related to increased storage of ice on land, but the cooling is only part of the story. 95 – tectonic stability - This is a critically important point - the question will be how this may relate stability back to MIS5. 103 - How deep can LIDAR penetrate into the water column? I presume the water is quite clear, but how much does it vary along the coast? 129 – see previous note: how shallow? 141 and throughout: “Fm.” punctuation for an abbreviation. 162 - T2 appears to be at -32 m on profile CC' 164 - T4 seems to be deepest at -54 m on profile AA', and very different in character from the other profiles. 172 – Be consistent in labeling profiles (not “Profilo”). 202 - ordering of references? I haven't see 38 and 39 yet (not until line 210). 269 - I did not see reference 47 used in the text. 271 – Figure 9 – This is a key part of the manuscript – increase the size of the figure to the maximum column width.

Author Response

Thank you for your referral.

Reviewer 3 Report

The manuscript deserves to be published - it has been built on results obtained from the use of sophisticated technologies and few coastal areas have been surveyed with such great detail in terms of geomorphological mapping.
Nevertheless provided results are not always supported by consistent data analysis and the terminology is not always correctly used through the paper. I report some comments within the attached pdf, but I would suggest being more consistent with the use of "landforms" "bedforms" "outcrop" and in general paying attention when you are describing a "morphology" and when you are giving evidence that clearly can indicate the processes that originated the mentioned morphology... In particular, I would add more indication confirming that the described terraces are erosional marine terraces - from their morphology they are undoubtedly terraces, but some evidence must be reported in order to support their description as erosional marine terraces (i.e. formed by past sea-level stand). The description of all submarine landforms is primarily based on their morphology, please check if there is any evidence to be reported from your survey that can clearly decipher the processes responsible for their formation. I also would include some literature that can give an indication regarding dominant drivers of seafloor geomorphic changes on the Apulian continental shelf. There should be some papers where the author can find evidence regarding Holocene and late Pleistocene sedimentation in the area. This could help in supporting the interpretation of terraces as marine erosional terraces.
4 profiles have been provided to show the position of the main detected terraces. But since there's a high-resolution DTM, I suggest checking if plotting the slope value (and the histogram of slope value) can help in detecting the depth of the more significant terraces. This should be a simple analysis (terrain analysis).
The conclusions basically list results from speculation regarding the correlation between sea-level fluctuations and the position of main terraces. I suggest including these sentences somewhere within the "result" section or within the "discussion" section since the correlation is not supported by absolute dating. Then the conclusion could be more concise and give more importance to the efficiency of the employed technique (the provided map gives, in any case, a lot of information that can be used to plan a more efficient survey in the future or sampling for dating).
Some parts of the manuscript seem to be well written, but others need to be edited regarding English language. I'm not a mother tongue, but some sentences did not sound correct to me.
Other detailed comments are reported within the pdf.

Author Response

Thank you for your revisions.

Round 2

Reviewer 2 Report

This version is much improved, with new and revised figures, and a much better text.  I am satisfied that this manuscript is ready to move forward.

Author Response

Manty thanks for your revision.